# Lipid Metabolism and Cancer

**DOI:** 10.3390/life12060784

**Published:** 2022-05-25

**Authors:** Hui Cheng, Meng Wang, Jingjing Su, Yueyue Li, Jiao Long, Jing Chu, Xinyu Wan, Yu Cao, Qinglin Li

**Affiliations:** Key Laboratory of Xinan Medicine, Ministry of Education, Anhui Province Key Laboratory of R&D of Traditional Chinese Medicine, Anhui University of Chinese Medicine, Hefei 230038, China; 2020201102004@stu.ahtcm.edu.cn (H.C.); wangmeng@stu.ahtcm.edu.cn (M.W.); sujingjing@ahtcm.edu.cn (J.S.); lyy18356406949@outlook.com (Y.L.); lj2892251508@outlook.com (J.L.); cj3974@outlook.com (J.C.); caolyn0725@163.com (X.W.); seattletszd@hotmail.com (Y.C.)

**Keywords:** lipid metabolism, cancer, programmed cell death

## Abstract

Lipid metabolism is involved in the regulation of numerous cellular processes, such as cell growth, proliferation, differentiation, survival, apoptosis, inflammation, movement, membrane homeostasis, chemotherapy response, and drug resistance. Reprogramming of lipid metabolism is a typical feature of malignant tumors. In a variety of cancers, fat uptake, storage and fat production are up-regulated, which in turn promotes the rapid growth, invasion, and migration of tumors. This paper systematically summarizes the key signal transduction pathways and molecules of lipid metabolism regulating tumors, and the role of lipid metabolism in programmed cell death. In conclusion, understanding the potential molecular mechanism of lipid metabolism and the functions of different lipid molecules may facilitate elucidating the mechanisms underlying the occurrence of cancer in order to discover new potential targets for the development of effective antitumor drugs.

## 1. Introduction

As an important nutrient needed for life, lipids are a general term for fats and lipids, providing energy and essential fatty acids that are required by the human body. Lipids are composed of thousands of different types of molecules, including fatty acids, phospholipids, sphingolipids, triglycerides, cholesterol, and cholesterol esters [1]. As the basic structure of cell membranes, lipids can provide the lipid components of biological membranes through energy supply, regulating cell membrane fluidity and lipid molecule signal transduction, and promote malignant biological behaviors, such as tumor cell proliferation, invasion, and metastasis.

Glycerophospholipids are the major skeleton component of the cell membrane. It regulates the fluidity of the cell membrane together with sphingolipids and cholesterol. Cardiolipin is a group of glycerol phospholipids with a unique structure, which consists of two phosphatidylglyceride skeletons and four acyl chains. Cardiolipin is considered to be the marker lipid of the mitochondria, which is critical for mitochondrial functions, including apoptosis signal transduction [2]. Ether lipids may be involved in cell membrane transport and cell signal transduction, which can be used as cell antioxidants and are commonly enriched in cancer cells [3]. Sphingolipids and fatty acids also play a role in signal transduction, while glycerides mainly serve as the major intermediates or lipid storage molecules in the synthesis of glycerol phospholipids. The increase in ceramide (CER), dihydroceramide (dhcer), and sphingosine (SPH) levels in cells is usually related to the induction of cell cycle arrest and/or cell death, while the increase in cer-1-phosphate, sph-1-phosphate, and two kinds of sugar sphingolipids, glycosylceramide and lactylceramide, levels is usually associated with the increase in cell survival, proliferation, migration, and invasion.

In the process of malignant transformation, tumor cells have malignant characteristics of infinite proliferation, invasion, and metastasis with a loss of control [4]. In addition, tumor cells are also characterized by an abnormally active glycolysis pathway, active glutamine metabolism, and abnormal lipid metabolism. Lipid metabolism is highly reprogrammed in cancer. In human cancer, fat synthesis is significantly up-regulated to meet the increased demand of membrane biosynthesis. Changes in lipid metabolism associated with cancer include increased lipogenesis and lipid uptake from the extracellular microenvironment, and enhanced lipid storage and mobilization from intracellular lipid droplets (LD) [5]. Changes in lipid composition and abundance, as the hallmark of cancer aggressiveness, may have a strong effect on the physical properties of cell membranes, such as lipid packing density, membrane fluidity or surface charge, and conversion into bioactive lipid mediators during signal transduction.

This paper addresses the processes of lipid synthesis, storage and metabolism, the key transduction pathways and targets of lipid metabolism, and the roles that different modes of lipid metabolism-mediated cell death play in tumorigenesis and progression.

## 2. Function of Lipids in Tumor Cells

As an indispensable class of biomolecules in the composition of cells, there is much less experimental research on lipids than on proteins and nucleic acids. Lipids, as highly complex and water-insoluble biomolecules, are essential for maintaining homeostasis, and tumor cells utilize lipid metabolism to support their rapid growth, migration, invasion, and metastasis. Lipids are mainly divided into two categories; one is triglyceride neutral fatty acids composed of one molecule of glycerol and three molecules of fatty acids, and the other category is lipids, mainly including phospholipids, cholesterol, and steroids. These substances hold a very important position at both the cellular and body levels. Lipids are derived from fatty acids (FA) that are an important component of triglyceride synthesis, which have the function of storing energy [6]. Phospholipids, one of the major types of membrane lipids, can also generate a variety of biologically active second messengers and promote the activation of signaling axes of normal and tumor cell genesis [7]. Finally, cholesterol can control membrane fluidity and the formation of microdomains, while sterols indirectly affect the production level of lipids in tumor cells by influencing sterol element-binding proteins [8] (Table 1). As the second messengers, lipids play an important role in signal transduction. A growing number of studies have indicated that tumor tissues require a high level of lipid metabolism to meet the increased functions of membrane synthesis, energy storage, and to transmit signals. The synthesis of fatty acids and the mevaleric acid pathway in the process of lipid metabolism are strongly associated with the growth, differentiation, migration, and invasion of cancer cells [9]. For example, the American Cancer Center found that the increase in lipid synthesis, storage, and metabolism occurs in a variety of cancers, and the reprogramming of lipid metabolism is an important marker for the diagnosis of malignant tumors. Because of the diversity and chemical complexity, the role of lipids in signaling, transcriptional regulation, and post-translational modification of proteins has attracted extensive attention. In summary, in order to have a deeper understanding the application of lipid metabolism processes in tumor treatment, it is particularly important to grasp the basic principles of lipid metabolism [4].

## 3. Synthesis and Storage of Lipids

### 3.1. Fatty Acids

Acetyl-CoA is a substrate for FA synthesis, which is produced from citrate or acetic hydrochloric acid; the main sources of citrate are glucose and glutamine, catalyzed by pyruvate dehydrogenase (PDH) and isocitric acid dehydrogenase (IDH), a process that takes place in the mitochondria. Secondly, acetyl-CoA is activated by acetyl-CoA carboxylase (ACC) to form malonyl-CoA, and acetyl and malonyl groups are coupled to the acyl carrier protein of multifunctional fatty acid synthase (FASN); subsequently, FASN catalyzes the formation of palmitate. Finally, under the action of stearoyl-CoA desaturase, fatty acid desaturase, and fatty acid elongation enzyme, a cell pool of non-essential fatty acids, including soft fatty acid, stearic acid, and oleate, is produced [11]. In addition to their role in fatty acid synthesis, lipids can also be ingested from the extracellular environment, i.e., exogenous uptake. For example, low-density lipoprotein receptor (LDLR)-mediated endocytosis is the method used by cells to obtain necessary FAs. Lipoprotein lipase (LPL), an extracellular lipolytic enzyme, is responsible for the hydrolysis of triglycerides in very-low-density lipoprotein (VLDL) and circulating chylomicrons. FA produced by LPL hydrolysis is transported into cells through fatty acid translocase (CD36) [12,13]. As a family of proteins involved in FA absorption and transport, fatty acid-binding proteins (FABPs) also contribute to the uptake of FA. The above-mentioned FA synthesis process in adults occurs mainly in liver, adipose tissue, breast, and intestinal epithelial cells [14].

FA is primarily stored in the form of triglycerides. Typically, the esterification reaction between glycerol with one equivalent of hydroxyl group and fatty acid with three equivalents of carboxyl group forms an ester bond with the glycerol backbone, thus producing triglycerides. FA addition to the glycerol backbone mainly relies on discrete acyltransferases [15].

Studies have shown that one of the early characteristics of tumor cells is the enhancement of FA biosynthesis, which is mainly manifested by the increased expression of ACC, FASN, CD36, and FABPs. Breast cancer pathophysiology suggested that the *ACC* gene is found in the recurrent amplicon of cancer, which leads to a reduction in the survival time of breast cancer patients. In tumor cells, 90% of fat synthesis source from the catalysis of FASN, and the proliferation of cancer cells requires the synthesis of biofilms, so it is no wonder that lipid synthesis is metabolically abnormal in tumor cells. In the process of lipid synthesis, inhibiting the activity of various enzymes can effectively reduce the synthesis rate of FA and attenuate the growth and proliferation of tumor cells [16] (Figure 1).

### 3.2. Phospholipids 

Phospholipids are synthesized mainly in the endoplasmic reticulum and are divided into glycerol phospholipids and sphingomyelins according to their backbone structures. The two hydroxyl groups on the glycerol molecule in glycerophospholipid are esterified by fatty acid, the main chain is glycerol-3-phosphate, and the phosphate group can be esterified by small-molecule compounds of different structures to form a variety of phosphoglycerides. Sphingomyelin is a phospholipid that contains sphingosine or dihydrosphingosine without glycerol. The basic raw materials for the production of glycerol phospholipids are glucose, fat and amino acids [17]. There are two ways to synthesize glycerophospholipids. One is to synthesize glycerophospholipids through diglyceride pathway with phosphatidylcholine (PC) and phosphatidylethanolamine (PE) as substrates. The other is synthesized bycytidine diphosphate-diglyceride pathway with phosphatidylinositol (PL), phosphatidylserine (PS), and cardiolipin (CL) as substrates through reactions with cytidine diphosphate diacylglycerol (CDP-DAG) as the intermediate [2].

Phospholipids synthesized on the endoplasmic reticulum are transferred to various organelles by multiple transporters. The first transporter is a PC-specific transfer protein that facilitates the movement of PCs between membranes while protecting the synthesis of mitochondrial PCs. Horibata observed the continuous expression of PC-specific transfer proteins in C2C12 cells and human primary myoblasts by immunofluorescence and immunoelectron microscopy indicating that this protein plays an important role in skeletal myogenesis [18]. The second is nonspecific lipid transporter, also known as sterol carrier protein-2. This protein is involved in the transport of multiple substances within the cell, binding to and exchanging multiple lipids [19]. Finally, the third kind of transporter is the PI transfer protein. The protein consists of two subtypes, PITPα and PIPTβ, with both being a soluble cytoplasmic proteins capable of transporting PC, PI, and the like in vitro and in the membrane [20]. These three transfer proteins enable phospholipid transfer activities to be performed efficiently.

Phospholipids present in a variety of chemical structures and are indispensable in cell adhesion migration and apoptosis [21]. Changes in the composition, distribution and metabolism of phospholipids in cancer cells are important detection markers. Environmental stressors in the tumor microenvironment can alter the activity of “phospholipid transporters” and the ability of cells to maintain plasma membrane phospholipid asymmetries, leading to phospholipid transporter imbalances [22]. Exposure of phospholipids allows tumor cells to survive and metastasize to initiate a coagulation cascade, which stimulates the release of cancer growth-promoting factors. Therefore, the synthesis of phospholipids inside and outside cancer cells plays an important role in tumor progression (Figure 2).

### 3.3. Cholesterol 

Cholesterol is an important component of biofilms, and is also the precursor to the synthesis of bile, steroid hormones, vitamin D, and other physiologically active substances [23]. Cholesterol in the body is derived from both endogenous synthesis and exogenous intake. Endogenous cholesterol synthesis is mainly through the mevalonic acid pathway with acetyl CoA as the basic raw material. The biosynthesis of cholesterol is firstly acetyl-CoA condensation of two molecules. Mammalian 3-hydroxy-3-methylglutaryl (HMG)-CoA synthase (HMGCS1) then condenses the third acetyl-CoA molecule to form HMG-CoA. Mammalian 3-hydroxy-3-methylglutaryl (HMG)-CoA reductase (HMGCR) reduces HMG-CoA to mevalonate (MVA), which is further converted to aromatic pyrophosphate (FPP), FPP decarboxylation forms isoprene, six isoprenes can be condensed to form squaleneethen generate lanolin sterol via a cyclic reaction, and finally lanolin sterol is converted into cholesterol [24]. It is worth noting here that the synthesis of cholesterol is regulated by HMG-CoA reductase. Exogenous uptake is mainly via the absorption of cholesterol from food through NPC1L1 on the membrane of intestinal epithelial cells [25]. It is subsequently esterified by sterol O-acyltransferase (ACAT) and finally absorbed by the liver in the form of chylomicrons. Regardless of endogenous synthesis or exogenous uptake, the liver is the main source of cholesterol. In addition, cholesterol carried on low-density lipoprotein (LDL) in the blood also can be absorbed by low-density lipoprotein receptors (LDLRs) on cell membranes [26].

Cholesterol was found in gallstones as early as the 18th century. Presently, adipose tissue is still considered to be the largest reservoir of cholesterol, and 25% of the cholesterol in a normal individual is stored in adipose tissue [27]. With free cholesterol observed in adipose tissue, some scholars believe that fat-free cholesterol mainly exists around the cell membrane and as lipid droplets. However the distribution of cholesterol is not observed inside lipid droplets, and cholesterol attached around fat droplets accounts for 1/3 of the total cell cholesterol [28].

In glioblastoma, epidermal growth factor (EGFR) has been shown to up-regulate the expression of LDLR at the transcriptional level [29]. In prostate cancer cells, activation of phosphatidylinositol-3 kinase/protein kinase (PI3K/AKT) has been demonstrated to contribute to the uptake of exogenous LDL, required for tumor cell growth. In addition, elevated cholesterol biosynthesis in tumor cells from breast cancer patients also suggests that it could be involved in tumor progression. These findings emphasize that cholesterol uptake plays an important role in the proliferation of certain cancer cell types (Figure 3).

## 4. Lipid Metabolism and Tumors

The balance of body’s internal environment plays a critical role in the regulation of lipid metabolism. Numerous studies have shown that cancer cells undergo altered lipid metabolism, and tumor lipid metabolism reprogramming can affect the structure of tumor cell membrane and the availability of lipids, thereby accelerating the induction of apoptosis of tumor cells [30] (Table 2).

### 4.1. Fatty Acids Metabolism Regulates Tumor Cells

The first step in fatty acid metabolism is fat mobilization. Fatty acid- and glycerol-free fat stored in white fat cells is released in the presence of hormone-sensitive triglyceride lipase (HSL), adipose tissue triglyceride lipase (ATGL), and various proteins. Second, glycerol is converted to 3-phosphoglycerol, which is utilized by the liver via glycerol kinase. Finally, triglycerides undergoes β oxidation, the process which is also known as fatty acid oxidation (FAO) [31]. Fatty acids are activated into agipyl CoA with CoA-SH as the carrier, and then the fatty acid CoA enters the mitochondria and undergoes dehydrogenation, water addition, decomposition in the four parts of dehydrogenation and pyrolysis to produce acetyl CoA, FADH2 and NADH [32]. Oxidation of fatty acids is a significant source of ATP, a process that produces twice as much ATP as glucose does [33].

The reprogramming of lipid metabolism, the basic substances needed to maintain cell proliferation, division, and ATP, is common among cancer cells. Endogenous fatty acids, in particular, are often up-regulated in tumor cells, and there is growing interest in fatty acid synthesis (FAs) and the role of FAO in the metabolic reprogramming of cancer cells [34]. Fatty acids are substrates that produce lipid signaling molecules, and the abnormality of fatty acid metabolism is closely related to the growth and differentiation of cancer cells.

ATP citric acid lyase (ACLY) is one of the key enzymes that catalyze acetyl-CoA to participate in fatty acid metabolism. The energy generated in the process of catabolism and biosynthesis is linked by ACLY to ensure the normal life activities of cancer cells. Because ACLY is usually over-expressed in a variety of cancers, changing the citric acid binding site of the enzyme, indirect transcyclic citric acid-binding is one of the cancer treatment options, an ACLY inhibitor, such as SB-204990, produces a strong inhibitory effect on lung cancer, prostate cancer, and other mouse tumors [35]. In addition, functional polymorphisms of the *ACLY* gene also can be used as independent prognostic markers for predicting recurrence and survival in patients with rectal cancer [36]. ACLY can be also involved in the vivo metastasis of colon cancer, promoting the migration and invasion of colon cancer cells [37]. In various types of cancer, decreased ACLY expression reduces the viability of cancer cells and inhibits the proliferation, invasion, and migration of tumor cells.

Acetyl-CoA can be carboxylated into malonyl-CoA by acetyl-CoA carboxylase (ACC). ACC is a rate-limiting enzyme for fatty acid synthesis and two subtypes has been identified in mammals, ACC1 and ACC2. ACC1 contributes to the production of fats and is essential for lipid synthesis; ACC2 is mainly involved in lipid oxidation. When one loses its function, the other can compensate for the loss, so the level of malonyl-CoA decreases only when both ACC1 and ACC2 are both non-functional [38]. During tumor lipid metabolism, ACC is a key enzyme. Studies have shown that pancreatic cancer cell viability will also be affected when ACC is suppressed [39]. In clinical models of ACC, it was found that metabolism of non-small cell lung cancer (NSCLC) is mediated by ACC, and when the ACC inhibitor ND-646 is used, the growth of NSCLC is also inhibited [40]. Mimicking ACC phosphorylation or the ACC inhibitor ND-654 can also inhibit the production of nascent fatty acids in the liver, thereby inhibiting the proliferation and differentiation of liver cancer cells [41].

Fatty acid synthase (FASN) is a key metabolic enzyme that catalyzes the de novo synthesis of saturated fatty acids, typically one molecule of acetyl-CoA reacting with seven molecules of malonyl-CoA to form palmitate. Studies have found that FASN hyperactivity is common in human epithelial cell carcinoma or precancerous lesions, associated with a higher risk of cancer recurrence and death. FASN inhibition reduces FAS, induces malonyl-CoA accumulation, and inhibits CPT1-mediated FAO, leading to cell cycle arrest and apoptosis of tumor cells. FASN over-expression can be detected in cancers such as ovarian cancer, pancreatic cancer, and non-small cell lung cancer [42,43,44]. The FASN inhibitor TVB2640 has shown significant anti-tumor activity in cancer cell lines and xenograft models, but its clinical translation and application are limited due to its drug properties [45].

There are two subtypes of SCD, SCD1 is the key product of membrane phospholipids and cholesteryl esters. Therefore, SCD1 plays an important role in the proliferation, migration, and invasion of cancer cells and is a promising anticancer target; SCD5 mainly plays a role in the brain and pancreas [46]. SCD1 can accelerate the production of oleic acid, and oleic acid can stimulate the migration of metastatic breast cancer cells. Therefore, SCD1 over-expression can often be seen in breast cancer detection reports [47]. In ovarian cancer, SCD1 is expressed at a high level in a variety of ovarian cancer cells, while also protecting ovarian cancer cells from death [48]. In an ovarian cancer model, the SCD1 inhibitor BZ36 blocks SCD1 production and sensitizes ovarian cancer cells to ferroptosis-inducing agents and induces tumor cell apoptosis [49].

During the intake of exogenous fatty acids, over-expression of CD36 can be observed in cancers, such as breast cancer, stomach cancer, and ovarian cancer [50,51,52]. Lack of CD36 in the prostate tissue of mice can attenuate cancer uptake of FA organisms and reduce tumorigenesis. In breast cancer patients, CD36 expression increased significantly after anti-HER2 treatment. An anti-CD36 antibody exhibits significant anti-tumor activity in human melanoma cells [53]. Those cancer cell experiments show that CD36 plays a key role in altered tumor metabolism. CD36 can promote the synthesis and storage of FA, so inhibiting the production of CD36 or modifying the binding site of CD36 can promote the apoptosis of cancer cells.

### 4.2. Phospholipids Metabolism Regulates Tumor Cells

Glycerol phospholipids are mainly degraded by phospholipase; sphingomyelins are intermediate products of sphingomyelins, while sphingomyelins can be degraded by sphingomyelin, and the lipid membrane structure of sphingomyelins and cholesterol is called a liposome, which is of important significance in cancer [3]. It was found that the synthesis of phospholipids was up-regulated, which could be conducive to the rapid proliferation and differentiation of cancer cells to ensure the normal physiological activities of newborn cancer cells [54]. Alterations in phospholipid metabolism are one of the key bases for early diagnosis of gastric cancer, by assessing serum phospholipid levels in patients with gastric cancer as a potential biomarker for early diagnosis of gastric cancer [55]. The one most closely associated with cancer is a biprotein/lipid phosphatase (PTEN). PTEN often leads to the development of cancer due to mutation, deletion, or loss of activity due to promoter methylation silencing [56]. Ding et al. found that the *PTEN* gene is often missing from the chromosomes of prostate cancer cells when screening for chromatin modifiers [57]. Inositol phosphate is the primary substrate of PTEN and acts as an important second messenger that transmits activation signals from growth factor receptors to organelles. These molecules serve as highly specific binding platforms to recruit proteins of interest onto specific membranes. Phosphatidylinositol (3,4,5)-triphosphate (PIP3) is the most typical lipid of this type. The molecule is produced from PI3K, which responds to growth factor signaling and mediates the recruitment and activation of AKT [58]. If the intracellular PI3K mutates, the carcinogenic activity will be enhanced [59]. Studies of ovarian cancer cells have shown that PTEN expression is elevated while PI3K/AKT activity is inhibited to promote apoptosis and reduce cell proliferation [60]. The PTEN/PI3K/AKT axes is an important pathway to regulate apoptosis, metabolism, cell proliferation, growth, and other biological processes, which is highly correlated with tumors [61].

### 4.3. Cholesterol Metabolism Regulates Tumor Cells

Cholesterol is converted mainly into bile acids in the liver, and it is also a raw material for the adrenal cortex, testicles, cortisol, and androgens. In addition, cholesterol can also be oxidized to 7-dehydrocholesterol in the skin, which is then converted into vitamins by ultraviolet radiation. Cholesterol metabolism is an important checkpoint for immune homeostasis and cancer progression, and liver X receptors (LXRs) are important regulators of intracellular cholesterol and lipid homeostasis [62]. Cholesterol can promote cell proliferation, migration and invasion, as well as metabolic abnormalities, such as increased cholesterol intake and up-regulated synthesis level, so as to confer tumor development. HMGCR is a key rate limiting enzyme in cholesterol biosynthesis. It converts HMG CoA to valproic acid. HMGCR over-expression promotes the growth and metastasis of cancer cells, inhibiting this enzyme has been exploited as a strategy for developing a drug for treating cancer [63]. In the tumor microenvironment, reprogramming cholesterol metabolism inside and outside the cell can promote the development of colon cancers, breast cancer, and prostate cancer [64,65]. Statins are inhibitors of HMGCR, and in breast cancer patients treated with statins, the lack or weak expression of HMGCR indicates good clinical outcomes. The new cholesterol drug pitavastatin plays a role in colon cancer stem cells by influencing the metabolism of cholesterol, inducing apoptosis of colon cancer cells [66]. The International Breast Organization (BIG) organized a randomized, phase III, double-blind trial OF BIG 1-98. 8010 patients with breast cancer after menopause were detected and observed, and the cholesterol level and cholesterol-lowering drugs (CLM) were measured at the beginning, and the analysis data was collected every 6 months to 5.5 years, and it was concluded that cholesterol-lowering drugs may have the effect of preventing early breast cancer recurrence during the period of auxiliary endocrine therapy [67]. Jamnagerwalla collected data from 4974 male non-statin users with elevated PSA and negative baseline biopsies, using multivariate logistic regression analysis to conclude that high serum total cholesterol is strongly associated with the risk of diagnosing high-grade prostate cancer [68].

Overall, major breakthroughs are still needed to make in the study of lipid metabolism, because the complexity of lipids’ structure makes it difficult for researchers to fully explain the mechanisms of lipid synthesis, storage, and metabolism. In cancer treatment, although lipid metabolism reprogramming plays an important role, the inhibition of a certain lipid or individual enzyme is not as effective in cancer treatment as expected. Once these problems are addressed, lipid metabolism reprogramming could be a preferred option for the treatment of cancer.

**Table 2 life-12-00784-t002:** Representative targets of anticancer drugs in lipid metabolic pathways.

Author, Year	Target Protein	Action Site	Inhibiter	Type of Cancer	Reference
Wei, J. et al., 2019	ACLY	Catalyzes acetyl-CoA binding	SB-2049990	Lung cancer, Breast cancer	[35]
Lally, J. et al., 2019	ACC	Catalyzed acetyl-CoA carboxylation to Malonyl-CoA	ND-654	Non-small cell lung cancer, liver cancer	[41]
VincentB, B.M. et al., 2018	FASN	Catalytic synthesis of palmitate	TVB2640	Pancreatic cancer	[46]
Tesfay, L. et al., 2019	SCD1	Elongate palmitate	BZ36	Ovarian cancer	[49]
Jiang, M. et al., 2019	CD36	Exogenous intake of fatty acids	Anti-CD36 antibody	Stomach cancer	[51]
Bjarnadottir, O. et al., 2020	HMGCR	HMG-CoA is converted to valproic acid	Statins	Breast cancer	[63]

## 5. Regulation of Lipid Metabolism in the Tumor Microenvironment

Lipid metabolism comprises lipid biosynthesis and lipid degradation, such as fatty acid metabolism, triglyceride metabolism, and cholesterol metabolism [69]. A vast number of proteases [70], receptors [71], and transporters [72] have been identified to be involved in the process of lipid metabolism, and they are controlled by a variety of signaling pathways, such as the WNT-β-catenin signaling pathway, the dysregulation of which promotes tumor stem cell renewal, cell proliferation and differentiation, leading to the progression of malignant tumors [73]. The HIF1/2 signaling pathway, which mediates multiple protective mechanisms that work together to maintain cancer cell survival in hypoxia by reducing oxidative metabolism [74]. These signaling pathways form a complex and elaborate regulatory network to maintain the homeostasis of lipid metabolism in the body (Figure 4) (Table 3).

### 5.1. Sterol Regulatory Element-Binding Proteins (SREBPs)

SREBPs control the expression of genes important for lipid synthesis and uptake and are endoplasmic reticulum (ER)-bound transcription factors [75], and play an important role in lipid metabolism. SREBP-1a, SREBP-1c, and SREBP-2 are three isoforms of SREBPs that bind to activate more than 30 genes involved in the synthesis, and uptake of cholesterol, fatty acids, phospholipids and triglycerides, respectively [76]. SREBP-1 has two isoforms: SREBP-1a and SREBP-1c. SREBP-1a integrates gene regulatory effects of hormones, cytokines, nutrition, and metabolites as lipids, glucose, or cholesterol through phosphorylation by multiple mitogen-activated protein kinase (MAPK) cascades [77]. SREBP-1c, an adipogenesis-controlling transcription factor, is induced in response to nutritional excess and stimulates the conversion of glucose to fatty acids and triglycerides for energy storage [78]. The rapid proliferation of tumor cells requires a substantial supply of lipids, with fatty acids, and triglycerides providing sufficient energy substrates [79]. Lipid uptake, storage and lipogenesis promote rapid tumor growth [4]. According to a growing body of data, SREBPs integrate multiple cellular signals to control lipogenesis, which is expected to become a key target of tumor therapy [16,80].

#### 5.1.1. Fatty Acid Metabolism in the Tumor Microenvironment

SREBP-1 serves as the critical bridge between oncogenic signaling and fatty acid metabolism, and its regulation in cancer cells is equally complicated [81]. SREBP1, the key regulator of FA synthesis, is an inactive precursor bound to the endoplasmic reticulum (ER) membrane and activates genes involved in the biosynthesis of FAs, cholesterol, and triglycerides, where cancer cells can de novo synthesize almost all FAs to support rapid tumor growth [82]. It was found that long-chain acyl-CoA synthase 4 (ACSL4) is required for de novo synthesis of FA in hepatoma cells. SREBP1 is a downstream target of ACSL4 and is essential for ACSL4-mediated tumor growth and metastasis, c-Myc activates and cooperates with SREBP1 to induce genes involved in FA synthesis, ACSL4 up-regulates lipogenic enzymes through the c-Myc/SREBP1 signaling pathway to regulate lipid metabolism and provide new insights into the treatment strategy of hepatocellular carcinoma [83,84]. In addition, fatty acid oxidation holds great potential in the treatment of cancer. Medium-chain acyl coenzyme dehydrogenase (ACADM) is a functional component that promotes β-oxidation, and SREBP-1 is a negative transcriptional regulator of ACADM. Cav1 is an important factor involved in tumorigenesis and progression of many cancers, and Cav1 can regulate fatty acid metabolism by activating SREBP-1, thereby inhibiting ACADM in HCC [85].

#### 5.1.2. Cholesterol Metabolism in the Tumor Microenvironment

*SREBP-2* is a gene that regulates the expression of the genes involved in cholesterol homeostasis. Key enzymes regulated by SREBP-2, such as 3-hydroxy-3-methylglutaryl coenzyme A (HMG-CoA) reductase (HMGCR) and mevalonate kinase (MVK), as well as key pathways mediated by SREBP-2, such as p53 and the phosphatidylinositol-3 kinase (PI3K)/Akt signaling pathway, are involved in the progression of many cancers [86]. SREBP2 activates the mevalonate pathway in p53-deficient colon cancer cells, promoting ubiquinone synthesis and altering the metabolic activity of tumor cells [87]. Leptin has been shown to enhance breast cancer cells proliferation, migration, and invasion by up-regulating acetyl coenzyme A acetyltransferase 2 (ACAT2) via the PI3K/AKT/SREBP2 signaling pathway [88]. Previously, we believed that the SREBP1/FASN signaling pathway controls fatty acid synthesis in various cancers. Recently, more and more studies have reported that the SREBP1/FASN axis is also involved in cholesterol synthesis. In rectal cancer, the SREBP1/FASN signaling pathway is activated rapidly under radiation stimulation, leading to cholesterol accumulation, cell proliferation, and inhibition of apoptosis, and blocking the SREBP1/FASN pathway hinders cholesterol synthesis and accelerates radiation-induced colorectal cancer cell death, which could be a potential target for colorectal cancer therapy [89].

In summary, the activation of SREBPs plays a critical role in boosting lipogenesis and promoting the proliferation of cancer cells; therefore, targeting SREBPs may be a novel cancer therapeutic development strategy.

### 5.2. Peroxisome Proliferator-Activated Receptors (PPARs)

PPARs are members of the steroid receptor superfamily and are encoded by three independent genes, namely PPARα (NR1C1), PPARβ/δ (NR1C2), and PPARγ (NR1C3) [10]. Their function is primarily regulated by ligand binding, which causes structural changes that affect their interaction with coactivators or co-inhibitors, activating or inhibiting functions [90]. PPARα and PPARβ/δ are mainly related to energy burning, while PPARγ promotes energy storage by enhancing adipogenesis [91]. For example, in a hepatocyte-specific PPARα knockout mouse model, impaired hepatic and systemic fatty acid homeostasis has been observed, leading to hepatic steatosis during aging [92]. In contrast, PPARγ activation of adipocytes may induce apoptosis of large adipocytes in mouse models of subcutaneous and visceral deposition, leading to deposition of mature adipocytes in human subcutaneous fat [93]. In addition, PPARs form heterodimer with retinoic acid X receptor (RXR) to regulate the expression of downstream target genes in response to ligand binding, so as to treat a variety of metabolic syndrome. Studies have shown that PPARα regulates lipid metabolism in the liver, and abnormalities may lead to hepatic steatosis and hepatocellular carcinoma. PPARβ/δ mainly promotes fatty acid β-oxidation in extrahepatic organs, while PPARγ accumulates triacylglycerols in adipocytes [94]. Thus, the PPAR family plays a crucial role in energy homeostasis and lipid metabolism regulation and is a promising therapeutic target [95].

#### 5.2.1. Fatty Acid Metabolism in the Tumor Microenvironment

Macrophages (TAMs) play important roles in tumor growth, invasion, angiogenesis and metastasis, and receptor-interacting protein kinase 3 (RIPK3) is down-regulated in hepatocellular carcinoma (HCC)-associated macrophages. RIPK3 deletion reduced reactive oxygen species and significantly inhibited caspase1-mediated cleavage of PPARs. These effects activate PPARs, promote fatty acid metabolism, including fatty acid oxidation (FAO), and induce M2 polarization in the tumor microenvironment. Loss of RIPK3 reprograms fatty acid metabolism through the ROS-caspase1-PPAR pathway, providing an anti-tumor immune therapeutic benefit against HCC [96]. Fatty acid-binding protein 7 (FABP7) plays a crucial role in triple-negative breast cancer (TNBC). In FABP7 over-expressing cells, activation of the PPAR-α signaling pathway limits metabolism and leads to reduced cell survival in response to serum starvation, thereby significantly inhibiting the proliferation of cancer cells [97]. Similarly, PPARγ agonists play a significant role in fighting against tumors, among others, by activating various signaling pathways to cancer cells and tumor stem cells [98]. In bladder cancer, fatty acid metabolism is activated, and the PPARγ antagonist GW9662 reverses cell cycle arrest. Through a PPARγ-mediated pathway, etomoxir induces bladder cancer cell cycle arrest in the G0/G1 phase and alters gene expression related to fatty acid metabolism [99].

#### 5.2.2. Lipogenesis

The regulatory processes of PPARs are closely related to the function of tumor stem cells (CSCs). PPARs are engaged in the epithelial–mesenchymal transition (EMT) process and the regulation of the function of CSCs; the regulatory function of PPARs confers them the ability to regulate multiple targets involved in cancer, thus making them potential candidates for cancer therapy [100]. A study showed that two important negative regulators of p53, the homologous proteins MDM2 and MDMX. PPARα activity is essential for MDM2 and MDMX to promote ferroptosis, suggesting the MDM2-MDMX complex regulates lipids by altering PPARα activity [101]. Colon cancer carcinogenesis is due to the deletion of intestinal PPARα through increased DNA methyltransferase1 (DNMT1)-mediated methylation of P21 in mice, and protein arginine methyltransferase 6 (PRMT6)-mediated methylation of p27 in mice. Compared to non-tumor tissues, human colorectal tumor tissues display lower levels of PPARα mRNA and protein and higher levels of DNMT1 and PRMT6. Drugs that activate PPARα may be used in the prevention or treatment of colon cancer [102]. Additionally, we found that invariant natural killer T (iNKT) cells increase lipid biosynthesis upon activation, and that is promoted by PPARγ and PLZF synergistically through enhancing transcription of Srebf1. Among these lipids, cholesterol is abundantly produced from iNKT cells necessary for IFN-γ induction of lactic acid in the tumor microenvironment which reduces PPARγ expression in intra-tumoral iNKT cells, thereby reducing cholesterol synthesis and IFN-γ production, and restoring lipid synthesis by activating PPARγ and improving treatment efficacy on tumors [103].

In summary, PPARs have attracted attention for their use as potential therapeutic targets for various diseases, and clinical trials are underway to identify them as potential therapeutic targets for cancer [104].

### 5.3. Liver X Receptors (LXRs)

LXRs are considered to be cholesterol sensors. There are two isoforms of LXRs, LXRα (NR1/H3) and LXRβ (NR1H2), both of which bind to the retinoic acid X receptor (RXR) to form heterodimers that can be activated by LXR and RXR agonists and exert biological effects by up-regulating target genes associated with reverse cholesterol transport, conversion of cholesterol to bile acids and intestinal cholesterol absorption [105], which are responsible for transporting cholesterol from the outside of the cell to maintain cellular cholesterol homeostasis [106]. Numerous studies have shown that the LXRs pathway regulates lipid metabolism and inflammation through the induction and repression of target genes [107,108,109], providing new therapeutic perspectives for subsequent studies of diseases associated with dysregulated lipid metabolism.

Activation of LXRs produces an antitumor benefit, and LXR mediates the expression of matrix metalloproteinases 2 and 9 (MMP2 and MMP9) via nuclear factor-kappa (NFκB). Activation of LXR is linked to NFκB-dependent repression of MMPs in hepatocellular carcinoma (HCC) cell lines, and LXRs can act as biomarkers for the regulation of tumor behaviour [110]. Under normal conditions, when intracellular cholesterol levels are high, cells synthesize oxysterols and activate LXR transcription, driving cholesterol efflux and reducing cholesterol inward flow and synthesis [111]. During carcinogenesis, cellular cholesterol homeostasis is disrupted and some studies have reported that increased cholesterol efflux or decreased internal flow is associated with tumor cell proliferation and tumorigenesis. Activation of LXRs in breast cancer induces the expression of ABC transporters, which promote cholesterol efflux, and low-density lipoprotein degraders (IDOL), which reduce cholesterol inward flow [112]. 27-Hydroxycholesterol (27-HC), an important regulator of cholesterol and breast cancer pathogenesis, promotes proliferation and growth of breast cancer cells, however, 27-HC can lower intracellular cholesterol levels by activating LXRs [113] and controlling 27-HC synthesis to mitigate the effects of cholesterol on breast cancer can aid in the development of new cancer treatment strategies.

Among various cancer cell lines, activation of LXR leads to cell death and the LXR-IDOL-LDLR axis is a common targeting pathway in multiple tumor types. LXR agonists lead to IDOL-mediated degradation of the low-density lipoprotein receptor (LDLR) and increase the expression of the ABCA1 cholesterol efflux transporter, and this axis may be used as a potential drug target in multiple cancers, while the LXR/ABCA1 axis also emerges as a promising drug target in cancer therapy by reducing cholesterol levels [114]. Studies have shown that if tumor growth is inhibited, LXR agonists can be used in combination with inhibitors of the fatty acid synthesis pathway. For example, the use of vitamin D3 (VD3, skeletal triol) or LXR agonist (T0901317) alone significantly reduces cholesterol accumulation in MCF-7 cells while inducing ABCA1 protein and gene expression, and the combination of the two is more effective in reducing cholesterol levels and increasing ABCA1 protein and gene expression compared with each treatment alone [115]. In summary, LXR is a promising molecular target in disease treatment.

### 5.4. AMP-Activated Protein Kinase (AMPK)

AMPK is a serine/threonine-protein kinase that exists as a heterotrimer comprising catalytic subunits α (α1 and α2), regulatory subunits β (β1 and β2) and non-catalytic subunits γ (γ1, γ2 and γ3) [116], AMPK senses cellular energy status through the competitive binding of three adenine nucleotides AMP, ADP, and ATP to three sites in its γ subunit, each of which in turn regulates the activity of the AMPK kinase structural domain in its α subunit [117].

In mammals, AMPK is a metabolic sensor and a key energy-sensing switch [118], that controls cell growth and processes, including lipid and glucose metabolism and autophagy [119]. AMPK primarily promotes the catabolism of glucose, and fatty acids, while preventing the synthesis of protein, glycogen, and fatty acids. Lipid metabolism begins when fatty acids enter the cell, and activation of AMPK increases fatty acid oxidation, promotes fatty acid metabolism, and affects lipid synthesis [120]. Many studies have demonstrated that the AMPK signaling system has implications in animal and human health, such as obesity and obesity-related metabolic diseases, type 2 diabetes, cardiomyopathy and [121,122,123], in particular, the progression of various cancers is closely linked to AMPK activity [124].

AMPK plays a key role in lipid metabolism by directly phosphorylating proteins or regulating gene transcription through multiple pathways. A acetyl coenzyme carboxylase (ACC) is an enzyme involved in the fatty acid synthesis and oxidation pathway, and AMPK has a regulatory role on ACC activity [125]. In pancreatic cancer cells, ACC serves as an important target for CPI-613, and inhibition of AMPK-ACC signaling significantly attenuates CPI-613-induced apoptosis, which could facilitate depicting the interaction between the AMPK-ACC signalling axis on lipid metabolism and apoptosis [126]. In hepatocellular carcinoma studies, Sorafenib has been found to affect the expression of stearoyl coenzyme A desaturase 1 (SCD1). Sorafenib treatment inhibits ATP production, leading to activation of AMPK phosphorylation, which interferes with SCD1-mediated monounsaturated fatty acid synthesis via the ATP-AMPK-mTOR- SREBP1 signaling pathway, reducing cell survival and thus killing hepatocellular carcinoma cells [127]. In clear cell renal cell carcinoma (ccRCC), the expression of ECHS1 is decreased, resulting in inactivation of fatty acid (FA) oxidation and activation of nascent FA synthesis, inhibiting the expression of AMPK-promoted ECHS1 transcriptional activator GATA3, and inactivating the AMPK-GATA3-ECHS1 pathway Leads to reprogramming of fatty acid metabolism in ccRCC, providing a new therapeutic approach for ccRCC [128]. Another study suggests that lysine demethylase (KDM5B), a novel regulator of lipid metabolism reprogramming, reduced the protein levels of fatty acid synthase (FASN) and ATP citrate lyase (ACLY) and was up-regulated in breast cancer, and that activation of the AMPK signalling pathway was involved in KDM5B-mediated reprogramming of lipid metabolism in breast cancer cells, making it A new strategy for the treatment of breast cancer [129]. Thus, AMPK regulation of lipid metabolism is a promising target in the treatment and prevention of cancer.

### 5.5. MicroRNA and LncRNA

microRNAs are small, evolutionarily conserved non-coding RNA molecules (containing about 22 nucleotides in length) that can regulate gene expression at the post-transcriptional level [130]. Studies have shown that microRNAs play an important role in the regulation of lipid metabolism, including lipid synthesis, fatty acid oxidation and cholesterol efflux [131,132,133], and can directly or indirectly mediate the genetic fine-tuning of cancer metabolism. However, microRNAs function only at the post-transcriptional level. In the cytoplasm, lncRNAs act as key factors regulating lipid metabolism in tumors, attenuating mRNAs, regulating mRNA stability or translation, competing with microRNAs for binding to mRNAs, and can be processed into microRNAs. And it interacts with microRNAs in cancer metabolism to influence tumorigenesis and progression [134].

#### 5.5.1. microRNA

miRNAs including miR-122, miR-370 and miR-33 are involved in the regulation of cholesterol homeostasis, fatty acid metabolism, and adipogenesis [135]. Among them, miR-122 is involved in regulating lipid and cholesterol metabolism, and its down-regulation is usually associated with hepatocellular carcinoma (HCC) [136]. Data analysis revealed that glutaminase (GLS) expression was negatively correlated with miR-122 in primary human hepatocellular carcinoma, inhibition of GLS1 reduces oxidative stress and restores VLDL triglyceride output, thereby reducing hepatic steatosis [137], and up-regulation of GLS RNA was associated with advanced tumor grade [138]. SIRT6 and miR-122 regulate each other in the liver and control various metabolic functions in the liver, especially reverse regulation of fatty acid β-oxidation. In patients with hepatocellular carcinoma, miR-122 and SIRT6 negatively regulate each other and miR-122 over-expression or down-regulation leads to a decrease or induction of SIRT6 protein levels, respectively, and the SIRT6-miR-122 correlation may serve as a biomarker for liver cancer prognosis [139]. MiR-33 is a post-transcriptional regulator of genes that regulate cholesterol efflux and fatty acid oxidation and is involved in the treatment of disorders of lipid metabolism [140]. MiR-33 targets adenosine triphosphate-binding cassette transporter protein A1 (ABCA1), and antisense inhibition of miR-33 in mouse and human cell lines leads to up-regulation of ABCA1 expression and increased cholesterol efflux to control cholesterol homeostasis [141]. There is a link between in vitro sensitization of breast cancer (IBC) cells and high-density lipoprotein (HDL), with MiR-33a negatively regulating the ABCA1 transporter and reducing HDL-induced radiosensitivity in cancer cells [142]. Another study indicated that MiR-33a expression negatively correlates with the gastric cancer biomarker CA199. miR-33a directly targeted cyclin-dependent kinases 6 (CDK6), cell cycle protein D1 (CCND1), and serine/threonine kinase PIM-1, and expression was down-regulated in gastric cancer tissues and cell lines, thereby inhibiting gastric cancer cell proliferation [143].

Fatty acid synthase (FAS) and acyl-coenzyme carboxylase 1 (ACC1) regulate fatty acid and triglyceride biosynthesis. miR-370 affects lipid metabolism by down-regulating FAS and ACC1 and targeting the 3’ untranslated region (UTR) of cpt1-1 to down-regulate carnitinepalmityltransferase 1-(cpt1-) gene expression [144]. In bladder cancer (BC), miR-370 expression is down-regulated during BC development. SOX12, a direct target of miR-370, attenuates miR-370-mediated tumor suppression through up-regulation. miR-370 mediates fatty acid oxidation by targeting SOX12, limiting cancer cell proliferation, migration and invasion, and ultimately blocking tumor metastasis [145]. Meanwhile, breast cancer studies have identified WNK2 as a downstream target of MiR-370, which mainly exerts its oncogenic function through down-regulation of the tumor suppressor WNK2, thereby affecting cell proliferation and tumor growth [146].

#### 5.5.2. LncRNA

Abnormal lipid metabolism is a hallmark of malignant tumorigenesis. Firstly, lipid synthesis can provide raw materials for the continued division and proliferation of tumor cells [147]. It was found that there is a large amount of adipogenesis in pancreatic cancer, and lncRNASNHG16 plays a regulatory role in adipogenesis. miR-195 is a functional direct target of lncRNASNHG16, and SREBP2 is another direct target of miR-195. lncRNASNHG16 accelerates the development of pancreatic cancer and promotes adipogenesis by directly regulating the miR-195/SREBP2 axis [148]. Another study pointed out that abnormal PCA3 expression regulated cellular triglyceride and cholesterol levels, and lncRNAPCA3 promoted antimony-induced disruption of lipid metabolism in prostate cancer by targeting the miR-132-3P/SREBP1 signaling pathway [149]. In thyroid cancer (TC), lncRNASNHG7 is oncogenic, and SNHG7 requires long-chain acyl coenzyme A synthase 1 (ACSL1) to promote TC progression. miR-449a can bind both SNHG7 and ACSL1, and SNHG7 can act as a sponge for miR-449a, thus increasing ACSL1 in TC cells, so the SNHG7/miR449a/ACSL1 axis contributes to the proliferation and migration of TC cells [150]. In addition, the target gene acyl coenzyme synthase 4 (ACSL4), another member of the ACSL family, is a downstream target of miR-34a-5p and miR-204-5p in prostate cancer. lncRNANEAT1 inhibits the expression of miR-34a-5p and miR-204-5p, regulates ACSL4 by sponging miR-34a-5p and miR-204-5p, promotes doxorubicin resistance in PCa cells, and accelerates the progression of prostate cancer [151]. In addition, triglyceride TG stores fatty acids, which provide a large amount of energy for cancer and promote tumor proliferation [152]. Among them, triglyceride lipase (ATGL) is considered as a key enzyme for the release of FAs from TG. It was found that in hepatocellular carcinoma (HCC), ATGL and its products DAG and FFA were shown to be associated with lncRNANEAT1-mediated HCC cell growth. lncRNANEAT1 competitively binds miR-124-3p to regulate ATGL expression, and through miR-124-3p/ATGL/DAG+FFA/PPARα signaling pathway attenuated the growth of HCC cells and disrupted the lipolysis of HCC cells [153] (Figure 5).

In summary, lncRNAs and microRNAs influence the expression of metabolites by regulating the involvement in lipid synthesis, fatty acid metabolism, and triglyceride metabolism, thus affecting cancer development.

**Table 3 life-12-00784-t003:** Genes and characteristics associated with lipid metabolism.

Cancer Type	Gene	Signaling Pathway	Function	Model	Cell Lines	Author, Year	Reference
Hepatocellular carcinm	*ACSL4*	c-Myc/SREBP1	Regulate fatty acid metabolism	HCC tumor samples	Not mentioned	Chen et al., 2020	[84]
	*ACADM*	Cav1/SREBP-1	Regulate fatty acid metabolism	FVB/N mice	Hep3B, PLC/PRF/5, LM3, MHCC97L150, Huh7, HLE	Ma et al., 2021	[85]
	*RIPK3*	ROS/caspase1/PPAR	Promote fatty acid metabolism	C57BL/6 wild type (WT) mice	H22 cells	Wu et al., 2020	[96]
	*SIRT6*	SIRT6/miR-122	Regulation of fatty acid β oxidation	6-month-old male mice	Huh7	Elhanati et al., 2016	[139]
	*NEAT1*	miR-124-3p/ATGL/ DAG+FFA/ PPARα	Regulate triglyceride metabolism	Patients and tissue samples	L02, 293T, HepG2, Huh7, SK Hep-1 and HCCLM3	Liu et al., 2018	[153]
	*SCD1*	ATP/AMPK/ mTOR/ SREBP1	Regulate fatty acid metabolism	Not mentioned	Huh7.5, HepG2 and Bel-7402	Liu et al., 2019	[127]
Breast cancer	*KDM5B*	AMPK/ KDM5B	Fatty acid metabolism	Not mentioned	MCF7 and MDA-MB-231	Zhang et al., 2019	[129]
	*ACAT2*	PI3K/AKT/ SREBP2	Regulate cholesterol metabolism	breast tissues	MCF-7, T47D and BT474	Huang et al., 2017	[88]
	*ABCA1*	MiR-33a/ABCA1	Control cholesterol homeostasis	Not mentioned	SUM149 and SUM159	Wolfe et al., 2016	[142]
	*FABP7*	PPARα/ FABP7	Regulate fatty acid metabolism	Not mentioned	Hs578T, MCF7, MDA-MB-231, MDA-MB-435S, BT474	Kwong et al., 2018	[97]
Pancreatic cancer	*AMPK*	AMPK/ACC	Regulate fatty acid metabolism	Not mentioned	AsPC-1 and PANC-1	Gao et al., 2020	[126]
	*SNHG16*	miR-195/SREBP2	Regulaties lipogenesis	Not mentioned	HPDE6-C7, PANC-1, AsPC-1, BxPC-3, SW1990, HEK-293	Yu, et al., 2019	[148]
Clear cell renal cell carcinoma	*ECHS1*	AMPK/ GATA3/ ECHS1	Regulate fatty acid oxidation	6- to 8-month-old littermates	Human HEK293T, ACHN and 786-O cells	Qu et al., 2019	[128]
Thyroid cancer	*ACSL1*	SNHG7/miR449a/ACSL1	Regulate fatty acid metabolism	Not mentioned	FTC133, TPC1, BCPAP, 8505C, Nthy-ori-3-1 cell lines	Guo et al., 2020	[150]
Bladder cancer	*SOX12*	miR-370/SOX12	Regulate fatty acid	Not mentioned	the human BC cell lines 5637 and J82	Huang et al., 2019	[151]
Colorectal cancer	*FASN*	SREBP1/ FASN/CHOL	Regulate cholesterol synthesis	Not mentioned	HT-29 and HCT-8	Jin et al., 2021	[93]
Glioblastoma	*ABCA1*	EGFR/AKT/ SREBP-1/LDLR	Regulates cholesterol metabolism	Not mentioned	U87, U87-EGFRvIII, U87-EGFR, U87-EGFR-PTEN, A431, LN229, T98	Guo et al., 2011	[119]
Gastric cancer	*CDK6* *CCND1* *PIM-1*	miR33a/CDK6/CCND1	Regulation of cholesterol homeostasis	Not mentioned	The human gastric carcinoma cell	Wang et al., 2015	[148]
Prostate cancer	*PCA3*	miR-132-3P/SREBP1	Regulates triglyceride	prostate cancer	LNCaP cell	Guo et al., 2019	[154]

## 6. Lipid Metabolism and Programmed Cell Death

### 6.1. Lipid Metabolism and Tumor Cell Apoptosis

Apoptosis is the physiological state of cell death, which is usually regulated by endogenous and exogenous factors. Various bioactive molecules, such as fatty acids, cholesterol, and ceramide, are produced in the process of lipid metabolism. They affect cell survival and apoptosis by participating in the activation or regulation of different signaling pathways. In addition, some reactive oxygen species (ROS) and ROS-dependent lipid peroxidation products are also factors that promote apoptosis. These mediators can activate apoptosis through mitochondria, receptors, or endoplasmic reticulum stress-dependent pathways [154].

#### 6.1.1. Fatty Acids

Related studies have shown that long-chain fatty acids may induce apoptosis by enhancing lipid peroxidation, mainly by inhibiting the expression of B lymphoma-2 gene (Bcl-2) via phosphorylation and enhancing the activity of cytochrome P450; these long-chain fatty acids may have cytotoxic effects on tumor cells at the level of oncogene expression [155]. However, few studies have been conducted on long-chain fatty acids, on the contrary, there are many more studies on polyunsaturated fatty acids. Zhang et al. found that polyunsaturated fatty acids (PUFA) can kill tumors. Research has indicated that PUFA can induce apoptosis of colon cancer cells LoVo and RKO, and is mainly mediated via the mitochondrial pathway, characterized by features such as the production of ROS, the accumulation of intracellular Ca^2+^, the activation of caspase-9 and caspase-3, the decrease in ATP level, etc. These factors may induce tumor cell apoptosis [156]. In addition, according to relevant research, ω-3polyunsaturated fatty acids (ω3-PUFAs) and vitamin D3 (VD3) play a positive role in reducing the incidence rate of breast cancer. The combination of ω3-PUFAs and VD3 has a strong effect on the apoptosis of three subtypes of breast cancer cell lines [157]. Several lines of evidence have shown that ω3-PUFAs have the potential to prevent and treat a variety of cancers, and docosahexaenoic acid (DHA) is one of them. ω3-PUFAs has been reported that it can inhibit the growth of a variety of cancer cell lines, induce cell death via apoptosis or autophagy. Kim et al. showed that ω3-PUFAs can induce glioblastoma (GBM) cell death through apoptosis and autophagy; ω3-PUFAs are expected to be used as chemopreventive and therapeutic drugs for GBM [158]. Colon cancer cells are usually sensitive to ω3-PUFAs-induced pro-apoptotic effects. It has been reported that the down-regulation of cyclooxygenase (COX-2) may be an important mechanism underlying ω3-PUFAs-induced apoptosis in colon cancer cells. Because COX-2 is known to have the anti-apoptotic ability and over-expressed in colon cancer, the efficacy of ω3-PUFAs in inhibiting the expression of this enzyme seems to be the key to explaining the antitumor effect of these fatty acids in colon cancer [159]. In addition, DHA and eicosapentaenoic acid (EPA) have been reported to protect against colorectal cancer, which may be at least partially related to their pro-apoptotic activity. DHA and EPA have important pro-apoptotic effects on different molecular types of colorectal cancer cells but do not affect noncancer cells [160]. Research shows ω6-PUFAs-derived eicosanoids have been demonstrated to have pro-inflammatory and pro carcinogenic effects [161]. In obese individuals, ω6-PUFAs derived from the twenty alkanic acid levels were observed to increase the number of breast cancer cells, thereby stimulating the initiation, invasion, and metastasis of breast cancer, thereby stimulating the initiation, invasion, and metastasis of breast cancer [162]. Studies have shown that, in pancreatic cancer models, ω6-PUFAs have antitumor activity and inhibit the continuous growth of pancreatic tumors, ω6-PUFAs inhibits the proliferation of prostate cancer cells and regulates inflammatory IL-6 and TNF by affecting the production of cytokines-α [163]. In addition, ω6-PUFAs can promote the occurrence and development of malignant tumors; ω6-PUFAs depend on the catalysis of cyclooxygenase-2 (COX-2) and produce prostaglandin-2 (PGE2) in vivo, to stimulate cell apoptosis. The imbalance of the expression of b-lymphoma-2 (Bcl-2) protein leads to cell proliferation and apoptosis, and to promote the occurrence and development of tumors. PGE2 can also promote the degradation of the extracellular matrix, thus further promoting apoptosis, invasion, and metastasis of cancer cells [164]. In the tumor microenvironment, ω6-PUFAs can up-regulate the production of PGE2 colorectal cancer cells and promote the transformation of myeloid inhibitory cells (MDSC) into M2 macrophages [165]; hypoxia-inducible factor-1 secreted by M2 macrophages α (HIF-1 α) promote tumor invasion and metastasis by inducing the expression of COX-2 and PGE2 in stromal cells and tumor tissues [166]. Fatty acid synthase is a key enzyme involved in fatty acid synthesis and is highly expressed in numerous cancers. Studies have shown that the inhibition of FAS in breast cancer cells can lead to the accumulation of propane two coenzyme A, resulting in the up-regulation of carnitinepalmityltransferase-1 (CPT-1) and ceramide and the induction of apoptosis, and the production of apoptotic genes BNIP3, TRAIL, and DAPK2, resulting in the apoptosis of tumor cells [167]. TVB-3166 is a selective FAS inhibitor. Its selective and effective FAS inhibition can lead to the selective death of tumor cells, and it has no significant effect on normal cells and can inhibit the growth of xenograft tumors in vivo at a well-tolerated dose [168].

#### 6.1.2. Cholesterol

Cholesterol regulates cell signals through cholesterol–protein interaction, cholesterol–phospholipid interaction, and membrane dynamics. Elevated cholesterol levels can promote cell proliferation. On the contrary, decreased cholesterol levels can promote cell apoptosis [169]. Recent studies have shown that cholesterol is also involved in the process of apoptosis. Tumor cells mainly rely on a large quantity of cholesterol to form new cell membranes and continue cell signaling. Eliminating lipid or cholesterol metabolism can reduce the metastasis rate and induce apoptosis of, tumor cells. There are also common genetic factors contributing to both apoptosis and cholesterol metabolism. MicroRNAs (miRNAs) are regulators of signaling pathways and cell process [170]. In addition, the increase in cholesterol levels in prostate cancer cells has been found to be the consequence of cholesterol dysregulation. Recent studies have demonstrated that protein kinase B (Akt) and sterol response element-binding protein are the main factors regulating cholesterol biosynthesis and feedback regulation [171].

#### 6.1.3. Ceramide

Ceramide can induce the apoptosis of a variety of cells, including tumor cells [172]. It has been reported that ceramide can induce apoptosis in tumor cells through anti-radiation. Ceramide is an effective inducer of apoptosis. It can cascade with stress-activated protein kinase (SAPK) and transmit death signals from cell membrane to the nucleus [173]. Related studies have shown that ceramide can mediate tumor-induced dendritic cell apoptosis (DC), mainly by down-regulating the intracellular phosphatidylinositol kinase (PI3K) pathway [174].

#### 6.1.4. Phospholipids

Phospholipid signaling molecules and their specific membrane receptors play an important role in the regulation of apoptosis. Phospholipid signaling molecules interact with G protein coupled receptors of endothelial differentiation gene (Edg) family, PI3K/AKT, MAPK/ERK, and other signaling systems, thus affecting cell apoptosis.

##### PI3K/AKT Signal Pathway

Phosphatidylkallidinol 3-kinase (PI3K)/protein kinase B (AKT) signaling pathway is one of the important survival pathways of cells. It plays an important role in programmed apoptosis. It regulates nerve cell survival/apoptosis, autophagy, neurogenesis, nerve cell proliferation, and synaptic plasticity [175], and can block the initiation of programmed apoptosis. It has also been confirmed by studies that inhibiting the activation of PI3K/Akt pathway can induce programmed cell death and inhibit tumor growth [176]. PI3K/Akt signaling pathway is closely related to apoptosis. After PI3K/Akt signaling pathway is activated, it can act on NF-κB. MTOR, glycogen synthase kinase-3 (GSK-3) β, nitric oxide synthase (NOS), and other downstream targets to play an anti-apoptotic role. In addition, activating the PI3K/Akt pathway can also inhibit mitochondrial-mediated apoptosis pathway to play an anti-apoptotic role.

##### MAPK/ERK Signal Pathway

Mitogen activated protein kinase (MAPK)/extracellular signal regulated kinase (ERK) plays an important role in the process of apoptosis, and the mechanisms underlying apoptosis is complex. ① The Bcl-2 family is an inhibitor of apoptosis protein, which plays an important role in regulating apoptosis. Abnormal activation of ERK can up- regulate the expression of anti-apoptotic members of the Bcl-2 family (such as Bcl-2, Bcl-XL and Mcl-1), especially the expression of the Mcl-1 protein, activates transcription factors, reduce the permeability of mitochondrial membrane, block the release of apoptosis inducing factor (AIF), and interfere with the activity of apoptosis protein mediated by tumor necrosis factor receptor, so as to inhibit apoptosis and promote the survival of tumor cells. ② ERK can play an anti-apoptotic role by inhibiting the expression of pro-apoptotic proteins Bim and bad. ③ ERK signaling pathway can also directly or indirectly inhibit the activity of apoptotic terminal effector caspase-3, thus blocking the process of apoptosis induced by various stimuli [177].

#### 6.1.5. P53

P53 tumor suppressor gene has been considered the most common mutant gene in human cancer. It is generally believed that the ability of p53 to induce aging and programmed cell death is the basis of its tumor-suppressive function. However, p53 has many other functions, such as tumor inhibition, especially p53’s control over metabolism and iron peroxide-mediated cell death [178]. p53 tumor suppressor protein plays a key role in cell defense against tumorigenesis. For example, p53 coordinates various pathways and helps prevent cell transformation by inducing cell cycle arrest, DNA repair, and apoptosis [179].

### 6.2. Lipid Metabolism and Autophagy of Tumor Cells

Autophagy is a self-eating process that uses lysosomes to degrade damaged, degraded, or aging macromolecules. Lipid metabolism refers to the synthesis and degradation of lipids (such as triglycerides, steroids, and phospholipids) to produce energy or produce the structural components of the cell membrane. There are complex interactions between lipid metabolism (such as digestion, absorption, catabolism, biosynthesis, and peroxidation) and autophagy, resulting in the regulation of intracellular homeostasis, including cell survival and death [180]. Lipid metabolism is involved in the formation of autophagic membrane structures (such as phagosomes and autophagosomes) during stress. Autophagy is a conserved catabolic process that transfers intracellular proteins and organelles to lysosomes for degradation and recycling [181]. Evidence over the past few decades has shown that autophagy is involved in the determination of cell fate and plays a key role in regulating cell energy and nutritional storage. Lipid droplets (LDs) are the main form of lipid storage in organisms. The autophagic degradation process of LDs refers to lipophagy or macrophages. Lipid uptake and de novo adipogenesis in cancer cells can be enhanced. Excess lipids in cells do not exist in the form of non-esterified free fatty acids (FFAs). Therefore, cells store excess fatty acids and cholesterol in the form of neutral and inert biomolecules, such as sterols stored in the cell structure, which are called lipid droplets or LDS [182]. LDs is a lipid deposition surrounded by phospholipids in cells, and finally, a structural protein called perilipin (PLINs) is separated from hydrophilic cytoplasm [183]. Cancer cells are characterized by up-regulation of Fas and FA uptake, resulting in increased accumulation of LDs. For colorectal cancer [184], it is also associated with lung cancer [185]. LDs are, as a whole or partially, encapsulated in a double membrane autophagosome, and then the cargos are delivered to lysosomes for degradation. The mobilization of lipids extracted from LDs is mediated by lipolysis. The dynamic interaction among cytoplasmic lipase and inhibitor protein and pericellular lipid regulates the rate of lipolysis [186]. Cells meet their energy needs through lipolysis while excess lipids to prevent the increase of storage compromise. It has been suggested that LD proteins (PLINs) are an important part of LDs, and their lipolysis is a prerequisite for degradation. The degradation of PLINs has now been shown to be associated with lipolysis [187], both of which are induced under starvation conditions. Studies have shown that PLINs are targets for chaperone-mediated autophagy (CMA) degradation [188]. Hsc70 binds to plin2 and plin3 by recognizing typical pentapeptide motifs present in both proteins. Hsc70 and PLIN complexes bind to lysosomal-associated membrane protein 2a (lamp2a), resulting in their uptake and simultaneous degradation by lysosomes. Studies have shown that the blockade of chaperone-mediated autophagy (CMA) will reduce the association between cytoplasmic lipase, autophagy mechanism and LDS, thus reducing the degradation of LDs [188]. Lysosome-associated membrane protein type 2a (lamp2a) is a key protein in the CMA pathway. The accelerated degradation of lamp2a determines the loss of lysosomal membrane stability. Nutrient deprivation is also an activator of CMA, which can selectively degrade PLIN (such as plin2 and plin3), and promote the hydrolysis of LDs. Therefore, the crosstalk between cell proteolysis and lysis mechanisms can regulate the turnover of LDs. Autophagy has been reported to have both positive and negative effects on tumor progression [189]. Autophagy breaks down damaged proteins and organelles, such as mitochondria and peroxisomes, which is the potential source of ROS. Therefore, autophagy can protect cells from oxidative stress injury and chronic inflammatory response [190]. In the presence of intracellular stress, autophagy eliminates damaged and toxic cellular components to ensure intracellular homeostasis (Figure 6).

### 6.3. Lipid Metabolism and Ferroptosis

Cell membranes or organelle membranes are particularly vulnerable to lipid peroxidation because they all have more polyunsaturated fatty acids (PUFA). The production of iron dead PUFA derivatives requires two enzymes in the endoplasmic reticulum: acyl CoA synthase long chain family member 4 (ACSl4) and lysophosphatidylcholineacyltransferase3 (LPCAT3). LPCAT3 can cause lipid peroxidation mediated by lipoxygenase (LOX) family to produce toxic phospholipid hydroperoxide (PLOOH). In contrast, acyl CoA synthase long chain family member 3 (ACSL3) or stearoyl CoA desaturase (SCD/SCD1)—mediated production and activation of monounsaturated fatty acids (MUFA, such as octadecenoic acid) competitively inhibit PUFA related ferroptosis. In addition, in the process of lipid metabolism, phospholipid metabolism is related to ferroptosis. Iron-dependent cell death is characterized by phospholipid oxidative damage. Therefore, cell metabolism and phospholipid composition usually affect the sensitivity to ferroptosis. Glutathione peroxidase (GPx4), the key regulator of iron sagging, can effectively reduce oxidized phospholipids and inhibit the activation of arachidonic acid (AA) metabolic enzymes, which are involved in the process of phospholipid peroxidation. In addition, glutathione independent ferroptosis inhibitor protein 1 (FSP1) reduces coenzyme Q10 through NAD (P) H and inhibits lipid peroxidation to resist ferroptosis; Cholesterol is also thought to be associated with ferroptosis. Exogenous cholesterol hydroperoxide induces ferroptosis in a dose-dependent manner (Figure 7).

#### 6.3.1. MUFA and PUFA

Exogenous monounsaturated fatty acids (MUFAs), such as exogenous oleic acid (OA) and palmitoleic acid (POA), inhibit erastin- and RSL3-induced ferroptosis in cancer cells [191]. Mechanistically, MUFAs inhibit the accumulation of lipid ROS, particularly at the plasma membrane, and displace PUFAs from this location in the cell. Nevertheless, monitoring the balance between PUFAs and MUFAs can predict the sensitivity of cells to ferroptosis. Studies have shown that the production and activation of monounsaturated fatty acids (MUFA, such as octadecanoic acid) mediated by acyl CoA synthase long chain family member 3 (ACSL3) or stearoyl COA desaturase (SCD/SCD1) can competitively inhibit PUFA related ferroptosis.

The oxidation of polyunsaturated fatty acids plays a central role in driving iron Verticillium disease [192]. Studies have shown that polyunsaturated fatty acids are the lipids most prone to peroxidation in human fibrosarcoma cells treated with erastin. Cells treated with PUFAs can enhance erastin-induced siderosis, which can be inhibited by exchanging or deleting genes required for PUFAs activation and incorporation into phospholipids [193,194] ROS-induced lipid peroxidation is widely considered to be the biochemical basis for ferroptosis. Cell membranes or organelle membranes are particularly vulnerable to lipid peroxidation because they all have more polyunsaturated fatty acids (PUFA). The production of ferroptosis PUFA derivatives requires two enzymes in the endoplasmic reticulum: acyl CoA synthase long chain family member 4 (ACSl4) and LPCAT3. LPCAT3 can cause lipid peroxidation mediated by lipoxygenase (ALOX) family to produce toxic phospholipid hydroperoxide (PLOOH).

Lipid peroxide accumulation is the key to ferroptosis. Lipid metabolomics shows that polyunsaturated fatty acids, such as arachidonic acid (AA) or adrenic acid (AdA), are the lipids most prone to oxidation reaction in the process of ferroptosis, and are regulated by three-synthase. ACSl4 catalyzes AA or AdA to form rachidonyl coenzyme A and adrenoyl coenzyme A, then LPCAT3 esterifies it into phosphatidylethanolamine (PES) to form AA-PE and AdA-PE, and finally are oxidized into lipid peroxide by lipoxygenase (LOX). Research has shown that knockout or inhibition of the above-mentioned three synthase can inhibit the occurrence and development of ferroptosis [195].

##### ACSL4

ACSL4 belongs to the family of long-chain lipodyl-coenzymes. The function of long-chain lipoyl-coenzyme is to bind long-chain fatty acids to coenzyme A to form long-chain lipoacyl-CoA for the formation of the corresponding lipids, which is the upstream rate-limiting enzyme of LPCAT3 [196]. ACSL4 is different from other long-chain lipoyl-coenzyme family members in that which preferentially activates arachidonic acid (arachidonic acid, AA) and eicosapentaenoic acid to form long-chain polyunsaturated lipoyl-coenzyme A. Over-expression of ASCL4 would up-regulate intracellular long polyunsaturated lipoyl-CoA levels and increase the oxidized phospholipid content on the membrane surface, thereby increasing the sensitivity of tumor cells to ferroptosisinducers. In certain tumor cells, for example, kidney cancer and breast cancer cells, ACSL4 is highly expressed [197,198]. Research has shown ACSL4 is preferentially expressed in basal-like breast cancer cell lines, promoting ferroptosis, while it is often silenced in most luminal-type breast cancer cell lines that are resistant to ferroptosis, suggesting that ACSL4 determines ferroptosis sensitivity in breast cancer [194].

##### LPCAT3

LPCAT3 serves as the critical link in the lipid synthesis pathway. Its function is to bind long-chain fatty acids in long-chain lipoatyl-coenzyme A to phosphatidyle-thanolamine (PE). It has been shown that high expression of LPCAT3 up-regulates the production of unsaturated fatty acids in the plasma membrane, thereby increasing cell sensitivity to ferroptosis. Knockdown of LPCAT3 expression reduced the cell sensitivity to ferroptosis, and affected the killing effect of the ferroptosis inducer RSL3. At present, LPCAT3 mainly exists in ER [194]. Once localized on the cell membrane, those polyunsaturated fatty acids (PUFA) undergo peroxidation to mediate ferroptosis [199]. LPCAT3 maintains the systemic lipid homeostasis by regulating lipid absorption, lipoprotein secretion, and de novo fat formation [200]. Studies have shown that the absence of LPCAT3 leads to an extreme reduction in membrane arachidonate levels during ferroptosis [201]. In general, LPCAT3, as a ferroptosis reminder downstream of ACSl4, is likely to regulate arachidonic acid metabolism. Targeting ACSL4-LPCAT3 pathway seems to be a reasonable option for the treatment of ferroptosis resistance.

##### Lipoxygenase (LOX)

Lipoxygenase (LOX) is also an important factor affecting ferroptosis. LOX is a class of non-hemoglobin-containing ferrases that function as the peroxidized unsaturated fatty acid [202]. In the ferroptosis pathway, LOX, as a lipid peroxidation plasma membrane, promotes ferroptosis of tumor cells. LOX5, LOX12 and LOX15 are the main members of the LOX family. LOX12 and LOX15 can directly peroxide the phospholipids on the film to form phospholipid peroxide. Through the catalysis of ferrous ions, lipid peroxides induce a chain oxidation reaction with other lipids, resulting in cell ferroptosis. LOX5 forms phospholipid peroxide by peroxidation of phospholipids that have been pre-esterified by cytosolic phospholipase A2, resulting in cell ferroptosis [203].

##### ALOXs

ALOXs is a non-heme iron-containing dioxygenase that catalyzes the stereospecific insertion of oxygen into PUFAs. A human has six ALOX genes (namely ALOX5, ALOX12, ALOX12B, ALOX15, ALOX15bBand ALOX3), which are named according to the position number of the carbon of ALOXs oxidizing its PUFAs substrate [204]. Because different ALOX family members are expressed in different tissue with varied expression profiles, the role of ALOX in ferroptosis depends on the environment. Studies have shown that ALOX15 seems to be the key mediator for transforming oxidative stress into lipid peroxidation and iron sagging [205]. It can bind to phospholipid ethanolamine binding protein 1 (PEBP1), which can produce lipid peroxide and regulate RSL3-induced death of bronchial epithelial cells, renal epithelial cells and neurons. In particular, the selective peroxidation of Sn-2-ETE-PE by ALOX15 and ALOX15-PEBP1 complexes is converted to sn2-15HPETE-PE, which is essential for ferroptosis [206]. In addition, studies have indicated that the proferroptotic function of ALOX12 is activated by its acetylation. For example, RSL3 induced the protein expression and acetylation of ALOX12 protein, while the addition of sodium hydrogen sulfide reduced the acetylation and ferroptosis of ALOX12 in mouse myoblasts and skeletal muscle [207].

#### 6.3.2. Phospholipid Metabolism

Studies have shown that phospholipid metabolism is involved in ferroptosis. The form of iron-dependent cell death is characterized by phospholipid oxidative damage. Therefore, cell metabolism and phospholipid composition usually affect ferroptosis sensitivity. Glutathione peroxidase (GPX4), as a key regulator of ferroptosis, can effectively reduce oxidized phospholipids and inhibit the activation of arachidonic acid (AA) metabolic enzymes that contribute to phospholipid peroxidation. In addition, glutathione independent iron death inhibitor protein 1 (FSP1) reduces COQ10 by NAD (P) H and inhibits lipid peroxidation to resist iron death. In contrast, FSP1/COQ10/NAD (P) H exists as an independent parallel pathway, which can cooperate with GPx4 and glutathione (GSH) to inhibit phospholipid peroxidation (PLPO) and ferroptosis.

##### Glutathione Peroxidase 4 (GPX4)

GPx4 is a cysteine reductase with its activity requiring glutathione (GSH). The light chain subunit SLC7A11 (solute carrier family 7 members 11) and the heavy chain subunit SLC3A2 (solute carrier family 3 members 2) are important components of transport system XC-. Their cystine transporters, which are connected together by disulfide bonds, can transport cystine into cells [208]. Once cystine enters the body, it is transformed into GSH through a chemical reaction. GPX4 mainly reduces the oxides of fatty acids, phospholipids, and cholesterol to stable hydroxyl groups by consuming GSH to inhibit the chain oxidation of the cell membrane. GPX4 reduces the conversion of lipid peroxides to lipid alcohols through GSH, thus protecting cells from ferroptosis under normal conditions. Therefore, inactivation of GPX4 or depletion of GSH will cause extensive lipid peroxidation and lead to ferroptosis in some cells or tissues. In addition, GPX4, as an enzyme that reduces lipid peroxidation in biofilm [209], can convert GSH into oxidized glutathione disulfide (GSSG) to reduce lipid peroxidation (LPO) and maintain cellular redox homeostasis [210]. Once the XC-/GSH/GPx4 axis of the system is inhibited, extensive LPO will occur, resulting in iron sagging. For example, erastin is a typical ferroptosis inducer, which directly inhibits the XC-activity of the system, thus destroying the redox balance, leading to LPO accumulation and ferroptosis [211].

##### Ferroptosis Inhibitory Protein 1 (FSP1)

Ferroptosis inhibitor protein 1 (FSP1) is also known as mitosis inducing factor mitochondria associated 2 (AIFM2). In previous studies, the function of FSP1 has been found to participate in the release of cytochrome c from mitochondria as a class of apoptosis related proteins [212]. Recently, some studies have shown that FSP1 is a kind of ferroptosis inhibitory protein, which plays a role by mediating the reduction of lipids. There are two reduction mechanisms of FSP1. The first is that FSP1 can regenerate coenzyme Q10 (COQ10) with NAD (P) H thus inhibiting lipid peroxidation, and then resist ferroptosis. COQ10 is an integral part of the respiratory chain and an important antioxidant in cells. Up-regulation of COQ10 can significantly affect the level of intracellular oxidative free radicals [213] and inhibit the occurrence of iron-induced cell death. The second is that FSP1 can use its special protein structure at the N-end to enrich on the membrane, and use COQ10 as the raw material to reduce lipid peroxide [214], so as to inhibit the chain oxidation of lipids. Due to the excellent lipid reducing ability of FSP1, tumor cells with high expression level of FSP1 often have strong anti-ferroptosis ability.

##### COQ10 Antioxidant System

Coenzyme Q10 is an endogenous fat-soluble antioxidant, which possesses both the reducing and oxidizing forms. Coenzyme Q10 is a downstream product of mevalonate pathway and an important molecule for regulating cell iron sagging. The reduced coenzyme Q10 can prevent the harmful oxidation of protein, lipid and DNA [215]. In addition to the already-known COQ10 metabolic pathway, recent studies have also shown a new role of apoptosis inducing factor mitochondrial related 2 (AIFM2, also known as FSP1) in mediating the reduction of COQ10 production [214]. Under normal conditions, AIFM2 mainly retains in the mitochondria. AIFM2 plays a dual role in promoting apoptosis or inhibiting ferroptosis through different subcellular translocation in response to different cell death signals.

##### NADPH

NADPH is essential for ferroptosis, not only for replenishing glutathione (GSH)—and thioredoxin (TXN)—dependent systems, but also for the biosynthesis of valproate and de novo synthesis and extension of fatty acids. In fact, the basal level of NADPH is considered to be a biomarker of ferroptosis sensitivity in several cancer cell lines [216], while the consumption of NADPH by cytoplasmic NADPH phosphatase mesh1 can promote ferroptosis [217]. NADPH can be produced in many ways such as pentose phosphate pathway (PPP), NADH phosphorylation catalyzed by NAD kinase (NADK), and the conversion of isocitrate to A-KG by NADP dependent isocitrate dehydrogenase (IDH). While PPP enzymes glucose-6-phosphate dehydrogenase and phosphogluconate dehydrogenase positively regulate ferroptosis, NADK and IDH2 have opposite effects [218].

#### 6.3.3. Cholesterol

Based on the ubiquity of cholesterol in eukaryotic plasma membrane and its inherent sensitivity to oxidants, such as hydroxyl radicals, cholesterol is also considered to be involved in ferroptosis [219]. Exogenous cholesterol hydroperoxides induce ferroptosis in a dose-dependent manner [220], which is exacerbated by a decrease in GPx4 activity [221]. In fact, GPx4 is the only enzyme known to directly scavenge cholesterol hydroperoxide, and cells over-expressing GPx4 are highly resistant to oxidative ferroptosis induced by cholesterol hydrogen peroxide. However, whether cholesterol oxidation truly induces ferroptosis remains unclear [222], because ferroptosis caused by GPx4 inhibitor RSL3 is exacerbated by exogenous PUFAs rather than cholesterol.

## 7. Conclusions

In conclusion, lipids are closely related to the occurrence and development of tumors. The absorption, synthesis, transport and metabolism of lipids have a profound impact on the pathogenesis of tumors. In particular, the close relationship between lipid metabolism and programmed death of tumor cells provides a new insight into the diagnosis and treatment of tumors. Research needs to be conducted to further explore the molecular pathways related to tumor abnormal lipids in the tumor microenvironment and the intervention of drugs, and interfere with the impact of lipid metabolism of cells in the tumor microenvironment on tumor progression and treatment response, especially in the field of anti-tumor immunity and anti-angiogenesis. The metabolic mechanism provides a scientific foundation for the application of tumor energy reprogramming theory in clinic.

## Figures and Tables

**Figure 1 life-12-00784-f001:**
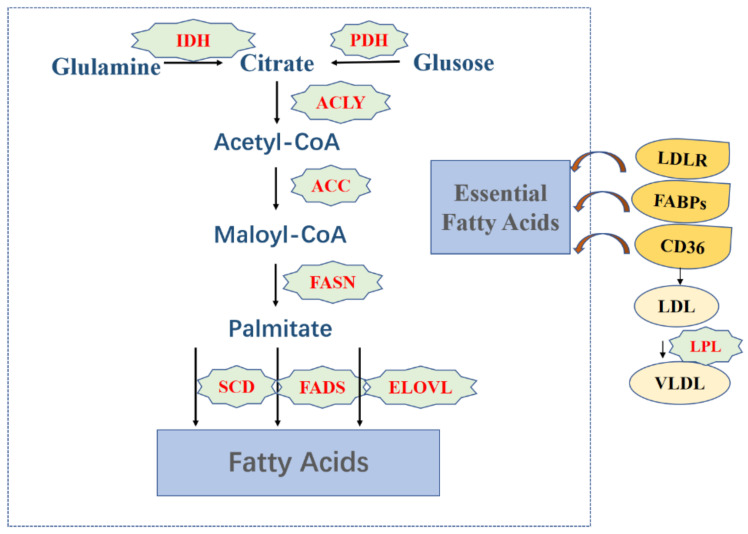
Fatty acid synthesis. Endogenous FA can be synthesized through two pathways. One is formed with glucose as the substrate under aerobic conditions; the other is formed with citrate as the substrate in the absence of oxygen. Citrate catalyzes the production of acetyl-CoA in ACLY, which is converted by ACC to malonyl-CoA. Acetyl-CoA and malonyl-CoA are FASN-catalyzed to synthesize palmitate. Finally, non-essential FAS cell pools are generated under the action of enzymes, such as SCD, FADS, and ELOVL. Exogenous uptake is a process to obtain extracellular FA via LDLR, FABPs, and CD36 by cells.

**Figure 2 life-12-00784-f002:**
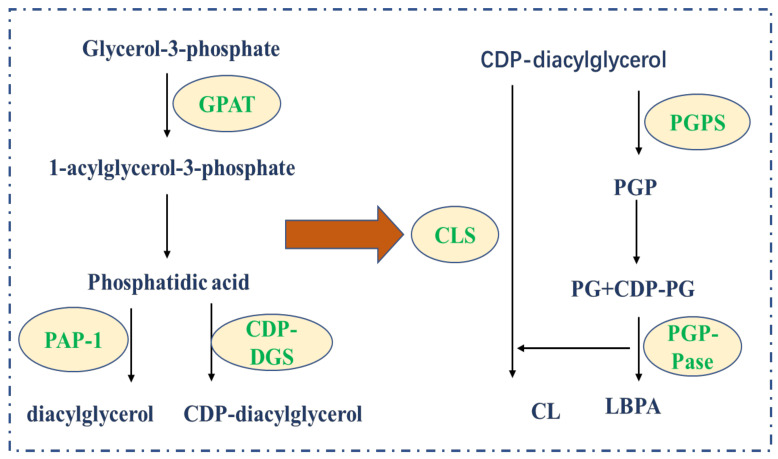
Synthesis of phospholipids. Firstly, glycerol-3-phosphate acyltransferase (GPAT) catalyzes glycerol-3-phosphate to form 1-acylglycerol-3-phosphate, and then converts to phosphatidic acid to produce diacylglycerols and CDP-diacylglycerolsby phospholipase (PAP-1) and CDP-DG synthase (CDP-DGS), respectively. CDP-diacylglycerol can be directly catalyzed by cardiophospholipid synthase (CLS) to form CL, or CDP-diacylglycerol phosphatase (PGPS) catalyzes CDP-diacylglycerol to generate phosphatidylglycerol phosphate (PGP) phosphatase to dephosphorylate PGP followed by reaction with cytidine diphosphate (CDP0) to form diphospholipidic acid.

**Figure 3 life-12-00784-f003:**
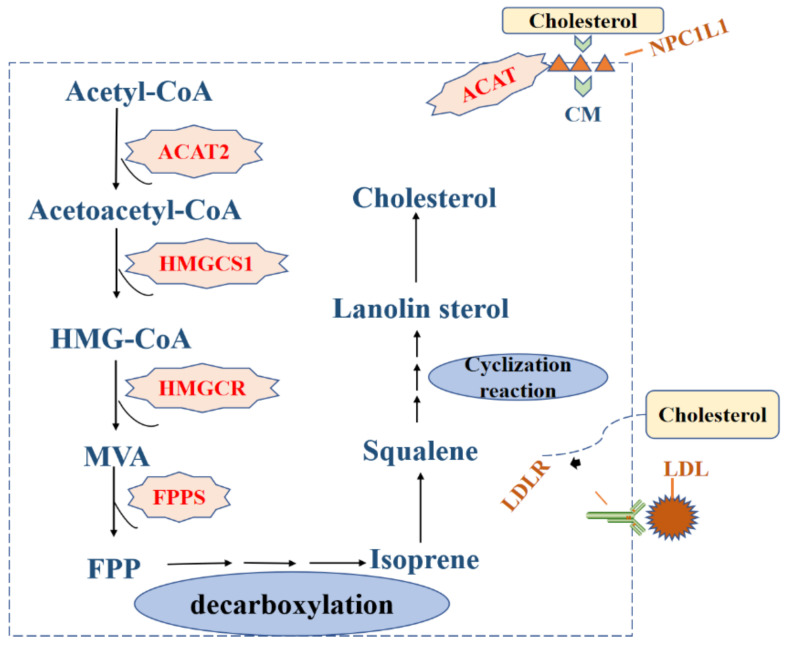
Cholesterol synthesis. In endogenous synthesis, two acetyl-CoA molecules are catalyzed by ACAT2 to form acetyl-CoA, followed by condensation of HMGCS1 and a third acetyl-CoA molecule to form HMG-CoA. HMGCR then reduces HMG-CoA to mevalonic acid, which is converted to pyrophosphate. Finally, oxidation produces squalene to produce cholesterol. Exogenous uptake: cholesterol in the blood and food is absorbed through LDLR and NPC1L1 on the intestinal epithelial cell membrane.

**Figure 4 life-12-00784-f004:**
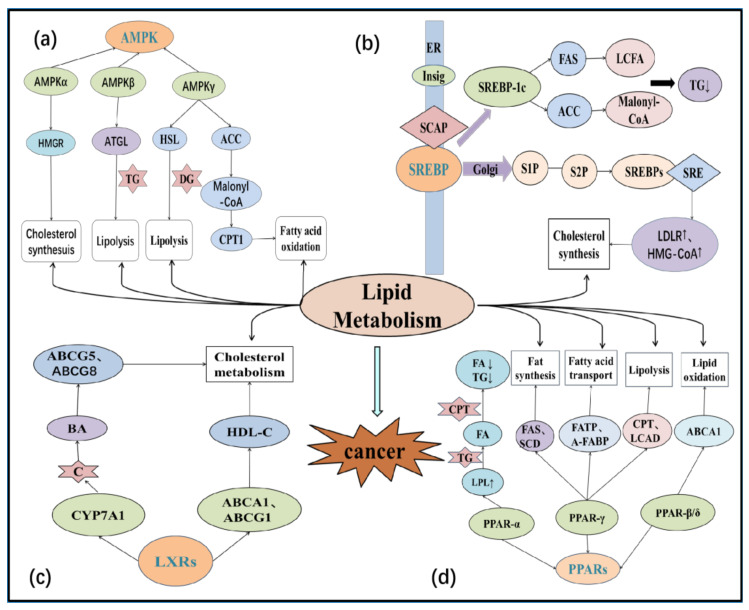
Signaling pathways involved in the regulation of lipid metabolism. (**a**) AMPK possesses three subunits, AMPKα, AMPKβ, and AMPKγ. HMGR can be phosphorylated by AMPKα to inhibit cholesterol synthesis; ATGL can be phosphorylated by AMPKβ to catalyze triacylglycerol hydrolysis and promote lipolysis; HSL can be phosphorylated by AMPKγ to catalyze diacylglycerol hydrolysis and promote lipolysis; ACC can be phosphorylated by AMPKγ to activate malonyl-CoA synthesis and inhibit CPT1 activity through negative feedback to inhibit fatty acid oxidation. (**b**) SREBP binds to SREBP cleavage-activated SCAP in the endoplasmic reticulum to form the SREBP-SCAP complex, which facilitates Insig separation at low concentrations and carries SREBP from the endoplasmic reticulum to the Golgi apparatus, followed by sequential cleavage at S1P and S2P to form a mature SREBP fragment, which binds to SRE in the regulatory region of lipogenic genes and enhances the expression of LDLR and HMG-CoA reductase genes together with relevant cofactors, thereby increasing endogenous cholesterol synthesis in cells. Among them, SREBP-1c regulates FASN to affect LCFA, and ACC to affect malonyl-CoA, thus inhibiting triglycerides. (**c**) LXRs can up-regulate the expression of CYP7A1, accelerate the conversion of cholesterol to bile acids and promote the excretion of ABCG5 and ABCG8 into bile and feces, thus affecting cholesterol metabolism; meanwhile, LXRs can induce the expression of ABCA1 and ABCG1, which transport cholesterol from cells to HDL molecules and promote cholesterol metabolism. (**d**) PPARs have three isoforms, PPARα, PPARγ, and PPARβ/δ. PPARα promotes lipoprotein lipase synthesis, catalyzes lipolysis of TG to free FA in lipoproteins, regulates FA transport to mitochondria through CPT expression, and reduces fatty acid and triglyceride synthesis; PPARγ regulates FAS, SCD thus promoting lipid synthesis, FATP, AFABP thus affecting fatty acid transport and CPT, LCAD thus affecting lipolysis; PPARβ/δ affects lipid oxidation by regulating ABCA1.

**Figure 5 life-12-00784-f005:**
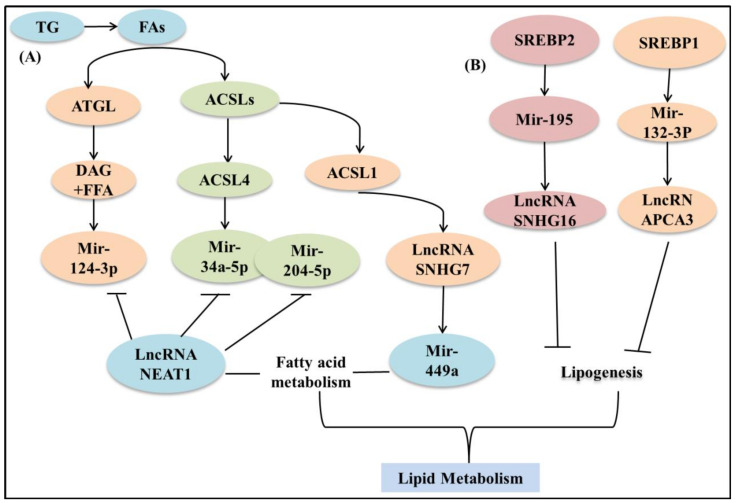
Regulation of lipid metabolism by LncRNAs. (**A**) FA can be stored as TG, and ATGL is the key enzyme for releasing FAs from TG storage; DAG and FFA are products of ATG, and LncRNANEAT1 competitively binds Mir-124-3p to inhibit its expression; ACSL4, a member of the ACSLs family, is a downstream target of Mir-34a-5p and Mir-204-5p, and LncRNANEAT1 inhibits its expression and affects fatty acid metabolism. (**B**) LncRNASNHG7 is the sponge of Mir-449a, which regulates ACSL1 expression. SREBP2 is a direct target of miR-195, which is a functional direct target of lncRNASNHG16, which inhibits lipogenesis by directly regulating its expression; LncRNAPCA3 is the mir-132-3P molecular sponge, and SREBP1 interacts with MIR-132-3P to inhibit cholesterol levels and affect lipid metabolism.

**Figure 6 life-12-00784-f006:**
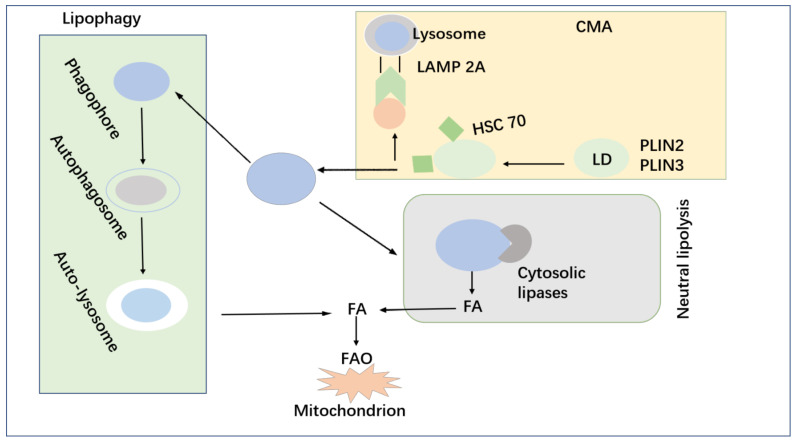
Lipid metabolism and autophagy. CMA (chaperone-mediated autophagy); FA (fatty acid); FAO (fatty acid oxidation); FFA (free fatty acid); Lamp (lysosome-associated membrane protein); LD (lipid drop); Hsc70 (heat shock homologous protein 71 kDa protein); lipid droplet surface proteins plin2 and plin3 are degraded by chaperone-mediated autophagy (CMA). The double membrane engulfs a portion of the whole LD to form autophagy bodies, which fuse with lysosomes to form autophagy lysosomes. Lysosomal acid lipase acts on lipids to form free fatty acids. Cytoplasmic lipase directly acts on the surface of LD and degrades lipids into fatty acids. Subsequently, fatty acid metabolism produces energy and metabolic intermediates via β- Oxidation in mitochondria. Lysosome-associated membrane protein type 2A (lamp2a) is a key protein in the CMA pathway. The accelerated degradation of lamp2a determines the loss of lysosomal membrane stability. It includes catabolism (fatty acid oxidation (FAO)), biosynthetic pathway (de novo fat formation), and storage of lipid droplets (LDS).

**Figure 7 life-12-00784-f007:**
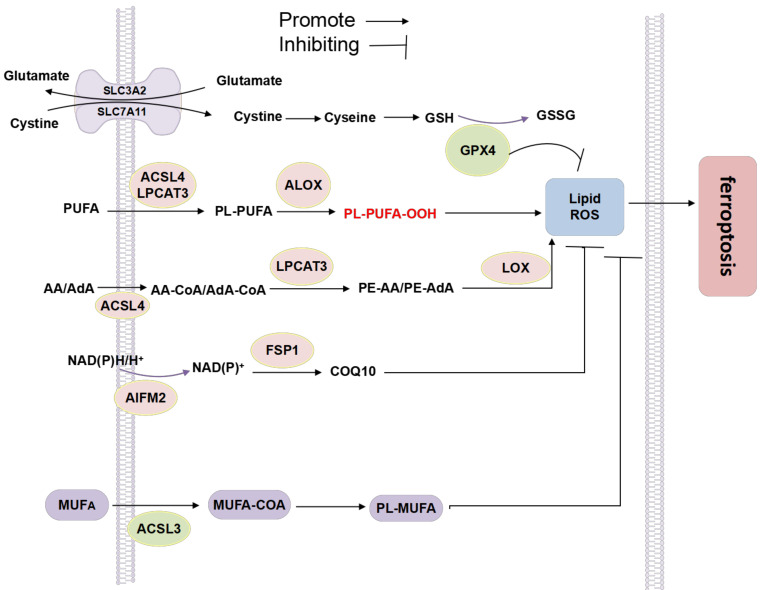
Lipid metabolism and ferroptosis. PUFA (polyunsaturated fatty acid); MUFA (monounsaturated fatty acid); Polyunsaturated fatty acids (PUFA); ACSL4 (acyl COA synthase long-chain family member 4); LPCAT3 (lysophosphatidyltransferase 3); AA (arachidonic acid); AdA (adrenic acid); AA CoA (arachidonic); AdA (adrenal coenzyme A); PE-AA (arachidonic acid phosphatidylethanolamine); PE-AdA (adrenic acid phosphatidylethanolamine); SLC7A11 (solute carrier family 7 member 11); Solute carrier family 3 member 2 (SLC3A2); COQ10 (coenzyme Q10). ACSL4 and LPCAT3 mediate the production of polyunsaturated fatty acids (PUFAs), which are essential for the induction of ferroptosis. In contrast, acyl-CoA synthetase long-chain 3 (ACSL3) and steroyl CoA desaturase (SCD) contribute to the synthesis of monounsaturated fatty acids (MUFAs), leading to ferroptosis resistance. Lipid peroxidation in ferroptosis. Arachidonate lipoxygenases (ALOXs) catalyze the stereospecific insertion of oxygen into PUFAs.

**Table 1 life-12-00784-t001:** Functions of different lipids.

Author Year	Function of Lipids	Type of Lipids	Reference
Cui, J. et al., 2020 Berger, J. et al., 2002	Engergy torage and metabolism	Triglycerides, diacylglycerin, Monoacylglycerol, long-chain fatty acids, sterol esters, PPARβ, PPARγ.	[6,10]
Dowhan, W. 2017 De Carvalho, C. & Caramujo, M. 2018	Signal transduction	Dicylglycero, arachidonic aicd, phosphatidic acid, ysophosphatidic acid, PI-4-phosphate.	[7,11]
Dowhan, W. 2017 Molendijk, J. et al., 2020	Membrane structure construction	SREBPs, LXRs, PC, PE, PL, PS, sphingomyelin	[2,7]

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
