# Peer review of "Lipid Metabolism and Cancer"

_life, 2022, doi:10.3390/life12060784_

Round 1

Reviewer 1 Report

The revised manuscript has been significantly improved. The authors only should check the Reference part again, like Ref.11, Ref.12 and Ref.85 are the same acticle.
And I  warrants publication in life now.

Author Response

Response to Reviewer #1’s Comments

The authors only should check the Reference part again, like Ref.11, Ref.12 and Ref.85 are the same acticle.

Response: Thank you for your careful check. We have revised the duplicate references. We feel sorry for our carelessness.

Reviewer 2 Report

This review by Cheng et al. under the title” Lipid metabolism and cancer” tries to explain the role of lipid metabolism in normal physiology and how that is changes during cancer and contribute to its progression.

concerns:

Although some of the write up had been revised this could still be improved

Instead of listing each gene list all around, please consider organizing the genes by shared pathways or organs/tissues.

In addition, a separate section focused on the different pathways and how more than one pathway converges and contribute to lipid signalling may be good.

A table collating all the genes shown to be involved in metabolism with their characteristics may be listed, such as animal, human, or in vitro, study design, sample size, etc

All these are still missing.

It also looks new authors are added now to the review.

Author Response

Response to Reviewer #2’s Comments:

Instead of listing each gene list all around, please consider organizing the genes by shared pathways or organs/tissues. In addition, a separate section focused on the different pathways and how more than one pathway converges and contribute to lipid signalling may be good. A table collating all the genes shown to be involved in metabolism with their characteristics may be listed, such as animal, human, or in vitro, study design, sample size, etc

Response: Thank you for your rigorous advice. Based on your comments, we have organised the genes in Part four according to cancer, checked the genes related to lipid metabolism and their properties and made a table, and integrated the contents of the original Table 3 in this one table. (Line698)

Reviewer 3 Report

Hui Chenga1, et al.

for “Life” review

This paper offers a deep revision of the metabolic paths in lipids vs. cancer metabolism. There are some concerns in reading the paper.

  • The paragraphs are headed by “lipids” ‘name, instead to introduce the audience towards the main aim of the revision which is the correlation with the carcinogenesis or metabolism in tumors.
  • “What is lipid” sounds as a book chapter in biochemistry and similarly the following paragraphs of the whole revision.
  • The Authors should share in the paragraphs between the lipid’s physiology and their role in tumors metabolism in a clearer way for audience.

Author Response

Response to Reviewer #3’s Comments

1.The paragraphs are headed by “lipids”‘name, instead to introduce the audience towards the main aim of the revision which is the correlation with the carcinogenesis or metabolism in tumors.

Response: We appreciate for your comment. In response to the review comments, we have revised the paragraph names to make them more directional. (Line62、Line234、Line309、Line335、Line371、Line417)

2.The Authors should share in the paragraphs between the lipid’s physiology and their role in tumors metabolism in a clearer way for audience.

Response: Thank you for your advice. We also inserted some roles of lipid synthesis and tumor between paragraphs to show the correlation between lipids and tumor, which facilitates readers to understand the relationship between lipid synthesis and tumor more clearly.(Line120-129、Line166-175、Line213-220)

Reviewer 4 Report

No comments

Author Response

Response to Reviewer #4’s Comments

Response: Thank you very much for your review of our article. 

Round 2

Reviewer 2 Report

The MS has improved.

Still there are typos and errors. Need to be checked for language

Author Response

 Thank you for your advice. We have asked experts to touch up the language throughout the article and to correct many errors.